# Impact of Obesity on Lung Function in Cats with Bronchoconstriction

**DOI:** 10.3390/vetsci9060278

**Published:** 2022-06-07

**Authors:** Alicia Caro-Vadillo, J. Alberto Montoya-Alonso, Laín García-Guasch

**Affiliations:** 1Internal Medicine and Animal Surgery, Faculty of Veterinary Medicine, University Complutense of Madrid, 28040 Madrid, Spain; aliciac@ucm.es; 2Internal Medicine, Faculty of Veterinary Medicine, Research Institute of Biomedical and Health Sciences (IUIBS), Universidad de Las Palmas de Gran Canaria, 35413 Las Palmas de Gran Canaria, Spain; 3IVC Evidensia Hospital Veterinari Molins, 08620 Sant Vicenç dels Horts, Spain; laingarcia@gmail.com

**Keywords:** obesity, cat, feline bronchial disease, chronic pulmonary disease, barometric whole-body plethysmography, lung function

## Abstract

Obesity is a nutritional disorder commonly diagnosed in adult cats that has been associated with an increased risk of different chronic diseases including respiratory diseases. The main objective of this study is to define if there is a relation between lung function measured by barometric whole-body plethysmography and obesity in cats with bronchoconstriction. Fifty-three cats were included in the study. All animals presented a bronchoconstriction status diagnosed with an Enhanced Pause (Penh) value higher than the reference range. Based on a standardized 9-point body condition scale, 36 cats were normal-weight cats (with BCS < 6), and 17 cats were considered overweight or obese cats (with BCS ≥ 6). Overweight cats were mainly male cats and older, and presented lower tidal volume values, lower minute volume values, and lower peak inspiratory and expiratory flows than normal-weight cats. According to the results of the present study, overweight cats showed a more compromised lung function parameters related to restrictive pattern compared with normal-weight cats. However, overweight cats did not show a higher bronchoconstriction level compared with normal-weight cats.

## 1. Introduction

Obesity is defined as an excess of body weight. This nutritional disorder is frequently diagnosed in cats and the reported prevalence of obesity in feline patients ranges from 11.5 to 63% [1,2,3]. The prevalence of obesity has been reported greater in older, neutered male cats than in females or unneutered males [1,2]. Other factors described as increasing the probability of obesity include breed, type of diet, feeding frequency, and the pet—owner relationship [3].

The ideal body weight of an adult cat ranges between 2 and 7 kg but it is preferable to use a standardized 9-point body condition scoring (BCS) system to classify pet cats, which not only considers weight but also the amount of fat in the animal [4,5]. Based on this scoring system, cats with BCS ≥ 6 and <7 are considered overweight, while cats with BCS ≥ 7 are considered obese [4,5,6,7].

In this sense, obesity has been described as a complex pathophysiological process associated with an increased risk of developing chronic diseases such as diabetes, cardiovascular diseases, obstructive sleep apnea, and certain cancers also in human medicine [8]. It is recognized that obesity predisposes to a variety of diseases such as lower urinary tract diseases, hepatic lipidosis, orthopedic diseases, skin and respiratory disorders, oncologic diseases, and endocrine diseases such as diabetes [1,4,7,8].

Asthma or eosinophilic bronchitis is one of the most prevalent pulmonary diseases in cats [9]. However, there are other feline chronic lower airway diseases that course with broncho-reactivity such as chronic bronchitis, bacterial bronchopneumonia, viral pneumonia, parasitism pneumonia, fungal pneumonia, intrathoracic neoplasia, *Mycoplasma* spp. bronchitis, etc. In cats, the most important clinical signs associated to all these diseases are related to bronchoconstriction (BC) and include cough, dyspnea, and tachypnea [10,11]. Barometric whole-body plethysmography (BWBP) is a noninvasive lung function test that allows to demonstrate a BC status in cats. It has been described as a useful way to diagnose BC [12,13,14], and to monitor the effectiveness of prescribed treatment [15].

Adipose tissue is an endocrine organ that produces numerous active mediators, such as cytokines and hormones which results in a low degree chronic inflammatory state. Some of these immune changes are also present in asthmatic and obese patients with respiratory diseases [7,8]. In a study performed with cats, the group of obese cats showed lower adiponectin concentration, higher plasma triglyceride, and serum amyloid A as inflammation markers [16]. Another study has proved the link between obesity and asthma in children and adult persons [8] This link has been shown to be secondary to dysanapsis, which induced airway hyperreactivity and increased levels of proinflammatory cytokines [8]. In addition, it has been shown that, in obese dogs, the respiratory rate increases whereas tidal volume (TV), inspiratory time (Ti), and expiratory time (Te) decrease [16]. In adult humans, overweight is associated with reduced lung function, forced expiratory volume and forced vital capacity are reduced, and obesity weakly affects total lung capacity [8]. Finally, decreases in functional respiratory capacity and expiratory reserved volume are usually associated with a restrictive lung function abnormality in obese patients [8].

In this context, the main objective of this study was to define if there is a relation between lung function, measured by BWBP, and obesity in cats suffering from BC, and the second objective was to demonstrate if obesity increases BC in cats.

## 2. Materials and Methods

Cats included in the present study were recruited from the Hospital Veterinari Molins IVC Evidensia (Barcelona, Spain) from January 20 to December 20. All patients were client-owned cats.

All cats underwent a complete physical examination (mucous membrane characteristics, cardiac auscultation, lung auscultation, arterial pulse characteristics, lymph nodes palpation, hydration, abdominal palpation, and temperature), radiographic examination (minimum two perpendicular views: lateral and ventrodorsal or dorsoventral views), as well as basic hematology (red blood cell count, white blood cell count, platelet count) and biochemical blood analysis (glucose, creatinine, urea, potassium, sodium, chloride, hepatic alanine transaminase) on day 0 (diagnosis day).

Inclusion criteria were defined as follows:Cats had to show clinical signs and/or radiographic findings compatible with feline chronic lower airway disease (chronic coughing, bronchial pattern, and/or interstitial pattern in radiographic studies);According to 9-point body condition score (BCS) scale [5], cats with BCS ≤ 6 were considered normal-weigh cats (NW-BC), while cats with BCS > 6 were considered overweight or obese cats (OW-BC);Barometric whole-body plethysmography parameters had to be compatible with BC, according to reference measures previously described [17] as the bronchoconstriction index Enhanced Pause (Penh) with a value higher than 0.460 ± 0.082 (normal range 0.205–0.539; according to our lab [17]);All the cats had to be dewormed within 3 months prior to the inclusion, and have negative results for feline leukemia virus, feline immunodeficiency virus, heartworm disease, and *Strongyloides* spp.;None of the cats included should have been treated with any drug for respiratory diseases such as corticoids or bronchodilators within 3 months prior to the inclusion. Cats treated with other medications, such as antimicrobial or antiviral drugs, were also not allowed in the study;None of the cats included could have a previous history of upper airway disease or pleural disease, and all of them should present normal cardiac auscultation and normal basic hematological and biochemical blood analysis.

Cats that did not fulfill the inclusion criteria were excluded from the study.

Ethical review and approval were not required for the animal study because it was conducted with client-owned animals, and no experimental animals were used. The study was carried out in accordance with the current Spanish and European legislation on animal protection (Royal Decree 53/2013 and 2010/63/UE Directive). Written informed consent was obtained from the owners for the participation of their animals in this study.

BWBP plethysmography was performed according to the technique described elsewhere [14] (Figure 1). The pulmonary function variables obtained were respiratory rate (RR (bpm)), tidal volume (TV (mL/kg)), minute volume (MV (mL/kg)), inspiratory (Ti (s)) and expiratory (Te (s)) intervals, Ti/Te ratio, bronchoconstriction index Enhanced Pause (Penh) and peak inspiratory and expiratory flows (PIF and PEF (mL/s)), mid-expiratory flow (EF50 (mL/s)) and peak to mid-expiratory flow ratio (PEF/EF50). In order to standardize TV, MV, PIF, and PEF according to BW, these parameters were divided by BW and renamed TV/BW, MV/BW, PIF/BW, and PEF/BW, respectively.

Data are expressed as mean ± SD. Comparisons were performed by the non-parametric Mann–Whitney U-test and Fisher’s exact test. A *p* < 0.05 was considered statistically significant. The statistical software SPSS v25.0 was used for this statistical study.

## 3. Results

A total of 53 cats were included in the present study. Thirty-six were NW-BC cats and 17 cats were included in the OW-BC group. The OW-BC group consisted of 15 male and 2 female cats, aged ranged from 1 to 15 years old, and weight range from 5.5 kg to 7.5 kg with a mean and a median of 6.5 kg. Four cats (23.5%) had a 7 out 9 of BCS, six (35.3%) had an 8 out 9 of BCS and the rest (7 cats, 41.2%) had a 9 out 9 of BCS. The vast majority were European domestic shorthair cats (*n* = 16) and just one was a Siamese. Thirty-six cats were included in the NW-BC group. Sixteen were males and 20 females. Their ages ranged from 1.2 to 14 years old, and weight range from 3.0 kg to 7.0 kg with a mean of 4.3 and a median of 4 kg. Breed distribution included European domestic shorthair (*n* = 31), Siamese and Sphinx with two representatives each, and one Radgoll cat (Table 1).

Statistically significant differences were observed between both groups in TV/BW, MV/BW, PIF/BW, PEF/BW, and PEFBW/EF50 ratio values. There were no other statistical differences between both groups of cats in the rest of pulmonary lung function variables: RR, Te and Ti intervals, Ti/Te ratio, EF50, and Penh (Table 2).

## 4. Discussion

The present paper provides the results of the BWBP data from a group of cats suffering from BC. To our knowledge, this is the first article to compare BWBP data in this particular group of cats.

The majority of cats included in the OW-BC group were neutered male of European domestic shorthair breed, and cats of this group were older than the NW-BC cats. This feature is in agreement with the data that describe obesity as more prevalent in male cats [2]. In the NW-BC group there were no sex differences, and this data agree with the knowledge of feline chronic disease [18].

OW-BC cats showed an RR similar to NW-BC cats. This feature is different from that described in obese dogs [19] and obese humans [20]. Obesity in dogs and humans produces a significant increase in RR to maintain the MV. However, in the present study there are no differences in the RR in cats with BC, as both groups of cats show a similar high respiratory frequency. This feature agrees with the results obtained in a previous study that compares lung function between healthy normal-weight cats and healthy obese cats [17]. It has been described that those cats with severe respiratory disease or cats with acute asthma attack, have a low RR. The BC produced by any bronchial disease strongly increases the Te interval, but the RR obtained from a patient without an acute state of BC is more variable and more dependent on the stress of the animal [21]. For this reason, it must be possible that the animals included in the present study, whatever the group, could be in a mild-to-moderate phase of chronic respiratory disease due to the fact that the RR interval did not differ between groups. It must be kept in mind that the physical stress due to the BWBP technique is the same for both groups. On the other hand, there has been reported a higher RR in young cats with respect to old cats [22]. However, although in the present study the mean age of NW-BC group is younger than in the OW-BC group, there are no statistically significant differences regarding RR.

The parameter TV/BW is lower in OW-BC cats, so this means that total lung capacity is also reduced. This finding is in agreement with that described in obese adult humans. Obesity affects the respiratory muscles such as diaphragm and intercostal muscles, producing an increase in RR and a reduction in the depth of respiration, which causes a lower TV [20]. In adult humans, these data are compatible with a restrictive lung pattern [8]. This pattern could be linked to the amount of fat that is accumulated into the thorax and abdominal cavity, which results in a limited movement of the diaphragm, an increase of pleural pressure and a decrease in compliance of the respiratory system [8]. This restrictive pattern is also associated with central obesity in human [23] which could be similar to the obesity in cats. Central obesity in human is more commonly seen in men than in women, as such, the restrictive pattern is more common in men [23]. In the present study a sex comparison has not been done because, although obesity is more frequent in males, fat distribution does not show sex differences in cats [2].

The MV is lower in OW-BC cats. There are some studies that have reported a reduction in lung volume and capacity in obese humans [8,20,23,24]. In fact, central obesity has been shown to be a restrictive factor associated with impaired lung function [23]. In a previous study performed with obese dogs, MV was not significantly different between groups due to RR [19]. However, in the present study, the RR do not vary, which means that the MV is lower in OW-BC cats with respect to NW-BC cats. It is necessary to keep in mind that BWBP reflects pulmonary mechanics and it is not a true assessment test [25].

The results of PIF and PEF are lower in OW-BC cats. It has been observed a 17% reduce in PIF and PEF values of OW-BC cats with respect to NW-BC cats. This observation is similar to the data reported in human medicine. Obese human patients have lower forced expiratory volume and lower functional residual capacity [8,20]. The finding of a lower PIF and PEF in OW-BC cats is also similar in obese dogs. Besides the amount of fat, other effects can affect ventilatory function such as compromised muscle strength, increased respiratory resistance and airway resistance [19]. On the other hand, there is a relationship between the level of physical activity and lung function. In this sense, a high level of activity means higher lung function [23], so obese cats with a low level of activity could have a worse lung function represented with lower PIF and PEF values.

There is no difference in the Penh value between groups in the present work. This is in disagreement with other studies that show a more severe bronchoconstriction pattern in obese children [8,26,27], but it is in agreement with the data observed by Manens et al. in obese dogs [19] and García-Guasch et al. in obese cats [17]. The first consideration is that Penh is calculated from pseudoflows by mean of BWBP, so it must be interpreted with caution. Second, the lung function parameters shown in children are associated with an obstructive pattern different from the restrictive pattern typical of adult humans [8]. So, for these reasons, the alterations in lung function parameters observed in the present study could be similar to the parameters observed in adult obese human patients, which are different from those observed in obese children.

Moreover, obesity causes mechanical dysfunction increasing resistance both in inspiration and expiration. As the Penh value depends on these two factors for its measurement, similar changes in both respiratory phases could result in a no different Penh value [28]. Other explanations could be the existence of airway dilatation due to a lower deep inspiration in obese animals [28].

On the other hand, it is well-known that adipose tissue produces proinflammatory cytokines as leptin, while there is a lower release of adiponectin (adipokines anti-inflammatory). Leptin increases the concentration of IL6, TNF-alfa and induces neutrophilic bronchial inflammation [28,29]. In this study, we do not know if the BC present in cats is related to eosinophilic or neutrophilic bronchial inflammation, regardless of the weight. This is another reason that could explain why there are no differences in BC measures such as Penh value.

## 5. Conclusions

In conclusion, OW-BC cats show more compromised lung function parameters related to restrictive pattern compared with NW-BC cats. However, OW-BC cats do not show a greater BC status compared with NW-BC cats.

These results are important from a clinical point of view, since OW-BC cats may show more difficulties in adapting to some circumstances that could influence lung function compared to NW-BC cats.

## Figures and Tables

**Figure 1 vetsci-09-00278-f001:**
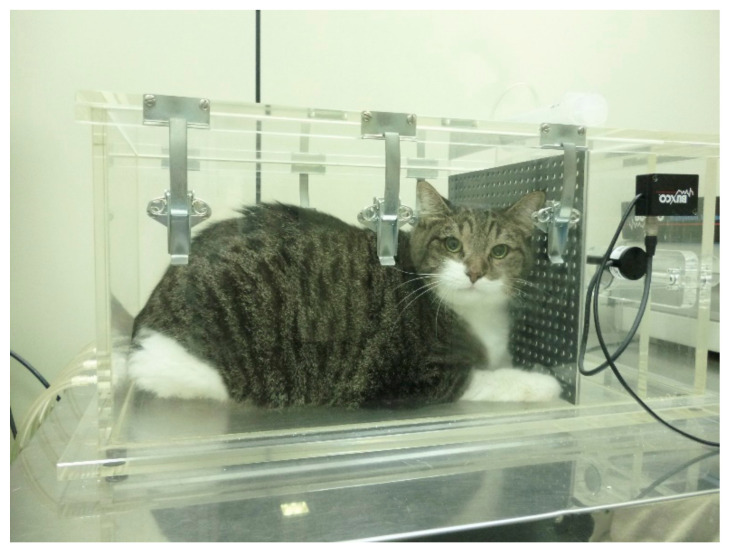
Obese cat placed in the barometric whole-body plethysmography chamber. The cat seems to be very comfortable during testing as the procedure is of limited or no stress to the patient.

**Table 1 vetsci-09-00278-t001:** Clinical data of the cats evaluated in this study. Data are expressed as mean ± standard deviation with maximum and minimum in brackets. It was observed that OW-BC cats were older and there were more male cats in this group with respect to NW-BC cats. The *p* value shows the statistical difference among OW-BC cats and NW-BC cats.

	All (53 Cats)	OW-BC (17 Cats)	NW-BC (36 Cats)	*p*
Age (years)	3.7 ± 0.35 (1–15)	7.3 ± 3.3 (1–15)	5 ± 3.6 (1.2–14)	0.019
Male/Female	31/22	15/2	16/20	0.003
Weight (kg)	4.74 ± 0.35 (3–7.5)	6.5 ± 0.47 (5.5–7.5)	4.3 ± 0.68 (3–7)	<0.001

**Table 2 vetsci-09-00278-t002:** Pulmonary lung function variables determined in each group. It was observed that all the variables with statistically significant differences had a lower value in OW-BC cats.

	NW-BC Cats	OW-BC Cats	*p*
RR (bpm)	69.6 ± 31.3	73.3 ± 28.3	0.481
TV/BW (mL/kg)	8.914 ± 4.334	6.453 ± 5.108	0.005
MV/BW (mL/m·kg)	495.925 ± 207.422	413.279 ± 287.138	0.016
Ti (s)	0.486 ± 0.192	0.421 ± 0.120	0.286
Te (s)	0.712 ± 0.311	0.581 ± 0.226	0.159
Te/Ti	1.458 ± 0.309	1.349 ± 0.245	0.245
PIF/BW (mL/s·kg)	29.312 ± 11.135	24.003 ± 17.572	0.006
PEF/BW (mL/s·kg)	28.672 ± 16.990	24.444 ± 25.216	0.008
EF50 (mL/s)	6.419 ± 3.458	8.733 ± 8.458	0.864
PEF-BW/EF50	4.647 ± 1.729	2.732 ± 0.549	0.0001
Penh	0.945 ± 0.418	0.922 ± 0.572	0.381

## Data Availability

The data presented in this study are available in the article.

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
