# Peer review of "Impact of Obesity on Lung Function in Cats with Bronchoconstriction"

_vetsci, 2022, doi:10.3390/vetsci9060278_

Round 1

Reviewer 1 Report

Impact of obesity on lung function in cats with bronchoconstriction

The study is straightforward and aims to define if there is a relation between lung function, measured by BWBP, and obesity in cats suffering from BC, and the second objective is to demonstrate if obesity increases BC in cats.

I do not have any major concerns and will recommend acceptance with the onus on the authors to address my very minor comments

·       I suggest organising some parts, such as the introduction and materials and method.

·       Cited references, especially those concerning obesity in cats, are quite old and more recent studies should be included.

·       The study mentioned that all the animals recruited in the study presented with bronchoconstriction status diagnosed with an Enhanced Pause (Penh) value higher than the reference range. Any other type of testing is used to diagnose these cats. And what is the reference range regards healthy and sick cats?

·       Line 30 in the introduction mentions the stat regards cat obesity but they don't support the statement with references

·       Line 32 needs clarification as the author mentions that the prevalence of obesity has been reported greater in older and neutered male cats than what exactly?

·       Line 36: the author mentions that The ideal body weight of an adult cat ranges between 2 and 7 kg but it is preferable to use a standardised 9-point body condition scoring (BCS) system to classify pet cats. It is a combination of the weight and the BSC

·       In line 43, the author mentioned that other studies had proved the link between obesity and asthma in 48 children and adults [6]. Can you clarify the exact meaning of this sentence and whether obesity acts as a predisposing factor for children's asthma?

·       Line 51 needs reference after the statement (. In the adult human, overweight is associated with reduced lung function, forced expiratory volume and forced vital capacity are reduced, and obesity weakly affects total lung capacity.) as not sure if it is connected to the sentence after this statement. If yes, try to rephrase the sentience to improve the clarity of the two sentences

·       Line 80 will help add the standard range for cat

(as the bronchoconstriction index Enhanced Pause (Penh) with a value higher than 0.465)

·       Line 85: Can you further clarify if that includes antimicrobial or antiviral treatment too? None of the cats included should have been treated with any drug for respiratory 85 diseases such as corticoids or bronchodilators within 3 months prior to the inclusion

·       The author needs to add the clinical history relevant to respiratory disease (how long the cat has been diagnosed, the severity of the condition, and any other treatment or care plan for the cat that is in place)

·       In line 87, the authors state that: None of the cats included could have a previous history of upper airway disease, and all of them should present normal cardiac auscultation and normal basic haematological and biochemical blood analysis.

However, the manuscript didn't list the previous clinical examination, any basic haematological and biochemical blood analysis values of the recruited cats. It would be helpful if the manuscript list all of these as well as the measurement frequency for pulmonary lung function and BWBP plethysmography during the study periods. Furthermore, did the study conduct a further clinical examination, and follow up basic haematological and biochemical blood analysis during the observation, if yes, can you add it to the manuscript.

·       In line 147, the author mention (a moderate state of BC) but has not defined previously what that exactly means

·       How can you determine this? For this reason, it must be possible that the animals included in the present study, whatever the group, could be in a mild-moderate phase of chronic respiratory disease.

·       Did the study measure any proinflammatory cytokines? If yes, can you add the results please to the manuscript as this will increase the quality of the current research.

·       please recheck if you are using the correct reference for the body score condition

Chiang, C.F.; Villaverde, C.; Chang, W.C.; Fascetti, A.J.; Larsen, J.A. Prevalence, risk factors, and disease associations of 229 overweight and obesity in cats that visited the Veterinary Medical Teaching Hospital at the University of California, Davis 230 from January 2006 to December 2015. Topics in Companion An Med 2022, 47:1-6

Author Response

We give thanks to the reviewer for the suggestion made for increase the quality of the article.

We have organized the introduction and materials and method as suggested.

We have introduced some recent studies concerning obesity in cats.

We haven`t used other type of testing to diagnose bronchoconstriction in cats. The reference range regards healthy cats for Enhanced pause is 0.389±0.127, according to the studies performed in our lab. And the obese cats (without bronchoconstriction or other respiratory diseases) showed Penh values over 0.460±0.082 (García-Guasch et al. 2019).

We have included references in line 30

We have clarified line 32

We have clarified line 36

We have clarified line 43

We have clarified line 51

We have included the standard range for Enhanced Pause (Penh) for cats in line 80

Line 85, we have included the suggestion.

We have included clinical history relevant to respiratory disease.

Line 87: we have included the list of clinical examination and haematological and biochemical blood analysis performed.

Line 147: We have rephrased the sentences to clarify it.

We haven’t measure any proinflammatory cytokines, although we considered this suggestion very useful for the future.

Reviewer 2 Report

The paper entitled “Impact of obesity on lung function in cats with bronchocon-striction” includes potentially relevant data for the veterinary health service directed to management of overweight/obesity directly and indirectly connected with pulmonary disorders. The Authors of this publication stated that overweight cats showed a more compromised lung function parameters related to restrictive pattern compared with normal-weight cats.

The Authors of this publication tend to conclude that overweight cats did not show a higher bronchoconstriction level compared with normal-weight cats.

Remarks:

The article lacks of the exclusion criteria.

The Authors did not present the biochemical characteristics of the animals included in the study.

The Authors of the manuscript did not – at least briefly – describe (in introduction section) the key risk factors for the development of obesity in the species of animals used in the study,

The Authors of this manuscript did not present (in the discussion section) a convincing attempt of the explanation of the obtained results - with regard to the comparable results obtained by other Authors.

The Conclusions section is laconic – it needs to rephrased.

The element of scientific novelty, the uniqueness of the described research results is difficult to point.

The Authors did not (clearly and convincingly) indicate/ describe the practical-clinical veterinary implications of the obtained results.

Author Response

We give thanks to the reviewer for the suggestion made for increase the quality of the article.

All the cats that did not follow the inclusion criteria, were excluded of the study. We have included this phrase in the text.

We have included the biochemical tests performed to the animals.

We have included the risk factors for the development of obesity in the cats.

We have tried to present convincing attempt of the explanation of the obtained results.

We have rephrased the conclusion section

We haven’t found any article similar so, we concluded this article is scientifically novel.

We have included a phrase to describe the practical and clinical veterinary implications of the obtained results.

Reviewer 3 Report

his is an interesting study that adds to the body of knowledge of respiratory disorders In obese animals. The authors provide appropriate human comparisons when appropriate; I suggest a rewrite of the sentence ending in line 42 for clarity. Several other minor English corrections are scattered throughout the text.

Author Response

Ok Thanks.

The manuscript has been reviewed by a specialized scientific translation service in the UK

Round 2

Reviewer 2 Report

The authors of the publication have modified the manuscript. The article was adapted to the reviewer's recommendations